# New Pyrazolo-Benzimidazole Mannich Bases with Antimicrobial and Antibiofilm Activities

**DOI:** 10.3390/antibiotics11081094

**Published:** 2022-08-12

**Authors:** Christina Zalaru, Florea Dumitrascu, Constantin Draghici, Isabela Tarcomnicu, Maria Marinescu, George Mihai Nitulescu, Rodica Tatia, Lucia Moldovan, Marcela Popa, Mariana Carmen Chifiriuc

**Affiliations:** 1Department of Organic Chemistry, Biochemistry and Catalysis, Faculty of Chemistry, University of Bucharest, 90-92 Panduri Road, 030018 Bucharest, Romania; 2“C.D. Nenitescu” Institute of Organic and Supramolecular Chemistry Romanian Academy, 202 B Spl. Independentei, 060023 Bucharest, Romania; 3National Institute for Infectious Diseases “Prof. Dr. Matei Balș”, No. 1 Dr. Calistrat Grozovici Street, 021105 Bucharest, Romania; 4Cytogenomic Medical Laboratory, 35 Calea Floreasca, 014462 Bucharest, Romania; 5Faculty of Pharmacy, “Carol Davila” University of Medicine and Pharmacy, Traian Vuia 6, 020956 Bucharest, Romania; 6Department of Cellular and Molecular Biology, National Institute of Research and Development for Biological Sciences, 296 Splaiul Independenţei, 060031 Bucharest, Romania; 7Department of Microbiology, Faculty of Biology, University of Bucharest, 1-3 Aleea Portocalelor St., 60101 Bucharest, Romania; 8Academy of Romanian Scientist, Ilfov No. 3, 050044 Bucharest, Romania

**Keywords:** pyrazoles, benzimidazoles, hybrid heterocyclic molecules, cytotoxicity, antimicrobial, biofilm formation

## Abstract

A new series of pyrazolo-benzimidazole hybrid Mannich bases were synthesized, characterized by ^1^H-NMR, ^13^C-NMR, IR, UV-Vis, MS, and elemental analysis. In vitro cytotoxicity of the new compounds studied on fibroblast cells showed that the newly synthesized pyrazolo-benzimidazole hybrid derivatives were noncytotoxic until the concentration of 1 μM and two compounds presented a high degree of biocompatibility. The antibacterial and antibiofilm activity of the newly synthesized compounds was assayed on Gram-positive *Staphylococcus aureus* ATCC25923, *Enterococcus faecalis* ATCC29212, and Gram-negative *Pseudomonas aeruginosa* ATCC27853, *Escherichia coli* ATCC25922 strains. All synthesized compounds **5a–g** are more active against all three tested bacterial strains *Staphylococcus aureus* ATCC25923, *Enterococcus faecalis* ATCC29212, and *Escherichia coli* ATCC25922 than reference drugs (Metronidazole, Nitrofurantoin), with the exception of compounds **5d** and **5g**, which are less active compared to Nitrofurantoin, and all synthesized compounds **5a–g** are more active against *Pseudomonas aeruginosa* ATCC27853 compared to reference drugs (Metronidazole, Nitrofurantoin). Compound **5f** showed the best activity against *Staphylococcus aureus* ATCC 25923, with a MIC of 150 μg/mL and has also inhibited the biofilm formed by all the bacterial strains, having an MBIC of 310 µg/mL compared to the reference drugs (Metronidazole, Nitrofurantoin).

## 1. Introduction

Pyrazoles are an important class of heterocyclic compounds and are promising scaffolds in medicinal chemistry. This class has attracted great attention due to its diverse pharmacological effects, especially anti-inflammatory, antimicrobial, antioxidant, anti-depressant, anti-influenza, and anticancer activity [1,2,3,4,5,6].

A number of drugs containing the pyrazole ring have been clinically tested and reported in the literature, including Lonazolac (a), a non-steroidal anti-inflammatory drug (NSAID); Diphenamizole (b), an NSAID and analgesic; Crizotinib (c), an anti-cancer drug acting as an inhibitor of ALK (anaplastic lymphoma kinase) and ROS; Pyrazomycin (d), an antibiotic, antineoplastic and antiviral drug (Figure 1) [7].

In turn, the benzimidazole ring has also emerged as an important heterocyclic system due to its wide range of biological activities, as well as synthetic applications in medicinal chemistry [8,9]. They are known for their crucial role in numerous diseases. Published literature has shown various substituted derivatives of benzimidazole nucleus exhibiting remarkable biological activities, such as antitumor, antiproliferative, anticancer, local anaesthetics [10,11,12,13,14,15,16,17,18,19,20,21,22] antimicrobial (including anti-HIV) [23,24,25,26,27,28,29,30,31,32], and antioxidant [33,34] effects. 

Rabeprazole (e) anti-ulcer; Astemizole (f), which presents an antihistaminic effect; Albendazole (g) with an antihelmintic effect; Bendamustine (h) an antineoplastic compound used in chemotherapy are some of the benzimidazole-core drugs (Figure 2) [35].

Moreover, benzimidazoles attached to other heterocyclic moieties have resulted in compounds (hybrid molecules) with improved anti-cancer [1,36,37,38] or antimicrobial action [30]. It appears that a combination of the two therapeutically active moieties, aryloxypyrazole and pyrido [1,2-a]benzimidazole, together in a single molecular framework may enhance the pharmacological activity of the title compounds [36]. So far, the literature has not presented many hybrid type pyrazolo-benzimidazoles [1,36,37,38].

Studies have also indicated that some Mannich bases possess anti-inflammatory, antibacterial, antimicrobial, antifungal, antihistaminic, and anti-tumor activity [39,40].

Previously our group successfully synthesized a novel series of alkylaminopyrazoles derivatives and most of the compounds showed enhanced antimicrobial activity [41]. In our previous study, we synthesized alkylaminopyrazoles with a suitable alkyl chain to have antimicrobial activity, and in addition, a supplementary pharmacophore pyrazole ring [41]. Starting from the previous results [41], we designed new molecules, by substituting the alkyl chain with another pharmacophore ring, resulting in a pyrazolo-benzimidazole hybrid, in order to obtain molecules with improved antimicrobial activity.

## 2. Resultsand Discussion

### 2.1. Chemistry

The literature mentions that Mannich bases still arouse interest to be studied due to the fact that they are potential medicinal agents [42]. The hydrogen atom of the imino group in the pyrazole ring is acidic enough to participate in the Mannich reaction. Therefore, Mannich bases can be obtained by multicomponent condensation between 1. a heterocyclic nucleus with the acid nitrogen atom (NH); 2. Formaldehyde; and 3. a secondary amine [43].

Unlike previous Mannich syntheses, which started from one amine, formaldehyde, and a carbonyl compound [44], in this case, the Mannich pyrazolo-benzimidazole **5a–g** bases resulted from the Mannich multicomponent condensation reaction between 1-aminobenzimidazoles, formaldehyde (30% aqueous solution) and pyrazoles.

Only pyrazole **1a** is commercially available. Pyrazoles **1b–d** were synthesized in good yields (79–89%) according to previously published methods [44,45,46,47] (Figure 1). Synthesis of 1-(hydroxymethyl)-pyrazole derivatives **2a–d** were realized in good yields (54–97%), as described by Dvoretzky and Richter [44]. Methylol compounds **2c** and **2d** were synthesized for the first time by our group, and are not described previously in the literature (Figure 1) [41,48].

The substituted benzimidazoles **3a,b** were obtained according to published methods [49] with yields of 75% and 48%, respectively, through the nucleophilic attack of 1,2-phenylenediamine on the carbonium ion intermediate, formed in an acidic medium from the corresponding carboxylic acid [50]. HOSA is an excellent reagent, used for the synthesis of a variety of nitrogen-containing heterocycles [51] in our laboratory (Figure 2).

The substituted 1-*H*-aminobenzimidazoles **4a,b** were obtained by the *N*-amination method with hydroxylamine-O-sulfonic acid (HOSA), under optimal reaction conditions of the substituted benzimidazoles **3a,b** with yields of 62% and 63%, respectively (Figure 2). HOSA reacts in basic conditions as a nucleophile, while in neutral and acidic conditions it functions as an electrophile [52,53]. 

The synthesis of the new substituted N-[(1*H*-pyrazol-1-yl)methyl]-1-amino-1*H*-benzimidazoles is summarized, and **5a–g** is shown in Figure 3. In the first step, pyrazoles **1a–d** reacted with an aqueous 30% formaldehyde solution to give the 1-(hydroxymethyl)pyrazole derivatives **2a–d**. In the second step, the 1-hydroxymethyl-pyrazoles **2a–d** reacted with 1-aminobenzimidazoles derivatives **4a,b** in the presence of methylene chloride at room temperature obtaining pyrazolo-benzimidazole Mannich bases **5a–g** in poor, moderate, to good yields 28–87% (Figure 3). The structures of all the compounds were established on the basis of spectroscopic data and microanalysis.

The first step in the Mannich reaction mechanism includes the formation of the iminium cation by the interaction between formaldehyde and amine in an acidic medium (Figure 4). The second step includes the nucleophilic attack of pyrazole nitrogen on the imine cation, with the formation of the final product.

The molecular formulas of the synthesized intermediate and final compounds were proved by elemental analysis and by the mass spectra of the target molecules.

The values of the R_f_ retention indices are given in the Experimental section.

#### 2.1.1. Spectroscopic Characterization of Compounds **5a–g**

##### IR Spectra

The IR spectra recorded in the 4000–400 cm^−1^ range in KBr pellets reflect the molecular structure of the new compounds and show the bands characteristic of secondary amines. In the IR spectra of compounds **5a–g** a medium band at 3266–3335 cm^−1^ was assigned to the stretching of ν_N-H_. The strong bands ν_Carom-N_ appear within 1166–1281 cm^−1^, and the strong bands ν_Caliph-N_ appear within 1034–1096 cm^−1^. The stretching bands due to the benzimidazole ring can be found at 1490–1557 cm^−1^, and 1342–1455 cm^−1^. The stretching bands due to the pyrazole ring can be found in the 1430–1489 cm^−1^ and 1306–1410 cm^−1^ ranges.

##### Electronic Spectra

The electronic spectra of the compounds were recorded in ethanolic solution and presented λ_max_ values in the characteristic range 268–328 nm of the chromophores in the molecule. In addition, these bands were assigned to the π-π* transitions.

##### NMR Spectral Analysis

The structures of the hybrid pyrazolo-benzimidazole compounds **5a–g** were confirmed by NMR spectroscopy. The formation of the pyrazolo-benzimidazole hybrids **5**, by condensation reaction between one molecule of substituted pyrazole and one 1-amino-1*H*-benzimidazole derivative was deduced from the value of the integrals of the two methylenic protons and the aromatic protons in the ^1^H-NMR spectra, as well as from the multiplicity of the protons in the CH_2_―NH moiety. The spin coupling between the protons in the methylene group (NHCH_2_) and the proton in the NH group is good evidence for the proposed structures. These protons appear usually as a doublet in the region 5.10–5.40 ppm with a coupling constant of ca. 5.5 Hz (in CDCl_3_), with few cases when they appear as a singlet and the NH proton shows a broad signal. The chemical shift of the NH proton was deduced by deuteration (D_2_O), being observed in the range 6.06–6.46 ppm. It is noteworthy that the NH proton appears strongly deshielded (δ_NH_ = 8.07 ppm) when DMSO-d_6_ is used as an NMR solvent. This is due to the hydrogen bond established between the amino proton and the oxygen atom of the DMSO-d_6_. The hydrogen atoms in the benzimidazole moiety appeared at the expected values and corresponding multiplicities. For compounds **5a–c** the H-2 from 1-amino-1*H*-benzimidazole ring is the most deshielded proton compared to those from the benzene ring as a consequence of the proximity between the two nitrogen atoms. The hydrogen atoms in the pyrazole ring presented expected values of the chemical shifts, determining the nature of the substituents.

The ^13^C-NMR spectra confirmed the structure of the compounds, which had been obtained in the reaction of 1-aminobenzimidazoles **4a,b** with *N*-hydroxymethylpyrazoles **2a–d**. The representative chemical shifts in the ^13^C-NMR spectra are those of the methylene groups (δ = 62.7–66.3 ppm). The chemical shifts for carbon atoms in the pyrazole and the benzimidazole rings appear in the expected ranges and are similar to those from the starting materials. The chemical shifts for the three carbon atoms in the pyrazole moiety are strongly influenced by the nature of substituents. For compounds **5a**, **5d**, and **5e** the C-4 in the pyrazole ring appears in the range 106.1–106.9 ppm. The presence of an iodine atom attached to C-4, as is the case in compounds **5c** and **5g**, induces a strong shielding of this carbon atom, which appears at 64 ppm. The carbon atom C-2 appears in the range 142.5–152.4 ppm, being deshielded by the direct influence of the two nitrogen atoms which are connected to it.

The recorded NMR spectra for the representative compounds are attached in the Appendix A.

##### MS Spectral Analysis

The base peak results from the cleavage of the C-N bond between the two nitrogen atoms, as shown in Figure 5 and Figure 6. This cleavage results in the base peak with m/z = 146, because the amine is not branched at “α” carbon. It is energetically favored by the breaking of the largest molecular fragment from the α carbon atom [54,55].

The newly synthesized compounds fragmented easily in the electrospray ion source. The fragmentation processes can support the structure formulas assigned to the new compounds. Figure 5 and Figure 6 show the specific fragmentation patterns. The characteristic ions are visible in the recorded spectra.

### 2.2. Biological Activity

#### 2.2.1. Evaluation of In Vitro Cytotoxicity of the Obtained Compounds

In our study, to analyze the cytotoxicity of pyrazolo-benzimidazole hybrid compounds **5a–g**, the L929 mouse fibroblast cell line was used for testing different concentrations of these compounds, ranging from 0.001 μM to 0.1 mM.

Cellular viability was evaluated by microscopic examination of cell morphology and by using the MTT assay. The results of the MTT test reflected the activity of mitochondrial enzymes, while the cell morphology allowed the observation of changes in the cell membrane, cytoplasm, and nucleus compared to the control of untreated cells.

Data shown in Table 1 revealed that the cytotoxicity of new compounds was dependent on the tested concentration. Results are presented as mean ± SD (n = 3). 

The fibroblast cell viability evaluation after 48h of cultivation in presence of pyrazolo-benzimidazole hybrid derivatives showed that all compounds were noncytotoxic in the range of 0.001–1 μM concentration. Cell viability at the concentration of 0.1mM ranged between 82.20–91.35% for compounds **5d**, **5e**, and **5a** which therefore proved to be biocompatible, while compounds **5b**, **5c**, **5f**, and **5g** were slightly cytotoxic, as suggested by the smaller viability values of 45.99–70.89%. At the highest tested concentration of 1mM very low viability values were recorded for all tested compounds, suggesting their high cytotoxicity at this concentration.

Figure 3 shows the cell viability of L929 fibroblasts cultured with pyrazolo-benzimidazole hybrid derivatives after 48h of experimentation evaluated by the MTT assay. 

Table 2 shows fibroblast morphology after 48h of treatment with samples (**5b**, **5c**, **5d**, and **5f**) in a range of 0.001 μM–1 mM concentration.

Cells treated with solutions of each compound showed normal morphology, similar to untreated cells, for the concentration range 0.001 μM–1 mM. In these cases, the density of the treated cells was comparable with the control culture. At a 1-mM concentration, the microscopy images of the cells treated with compounds **5b**, **5c**, **5d**, **5f**, and **5g** indicated a reduced cell proliferation (Table 2).

#### 2.2.2. Antimicrobial Activity

##### Activity against Planktonic Microbial Cells

The antibacterial activity against planktonic microbial cells of the new synthesized compounds was assayed on Gram-positive strains *Staphylococcus aureus* ATCC25923, *Enterococcus faecalis* ATCC29212 and Gram-negative strains *Pseudomonas aeruginosa* ATCC27853, *Escherichia coli* ATCC25922.

The reference drugs Metronidazole (a) and Nitrofurantoin (b) are shown in Figure 4.

The antibacterial activity of the obtained compounds against planktonic strains was quantified as the minimum inhibitory concentration MIC expressed in μg/mL (Table 3). Figure 5 shows the minimum inhibitory concentration of the tested compounds. 

In order to highlight the influence of the grafted substituent on the pyrazole and benzimidazole rings, the following conclusions were reached: all synthesized compounds, **5a–g**, are more active against all three tested bacterial strains *Staphylococcus aureus* ATCC25923, *Enterococcus faecalis* ATCC29212, and *Escherichia coli* ATCC25922 than the reference drugs (Metronidazole, Nitrofurantoin), with the exception of compounds **5d** and **5g**, which are less active compared to Nitrofurantoin. However, all synthesized compounds **5a–g** are more active against *Pseudomonas aeruginosa* ATCC27853 compared to the reference drugs (Metronidazole, Nitrofurantoin). Compound **5f**, which has the nitro group in position 4 of the pyrazole ring, and the benzimidazole ring is substituted in position 2 with the methyl group, showed the best activity against *Staphylococcus aureus* ATCC25923, with a MIC of 150 μg/mL compared to the reference drugs (Metronidazole, Nitrofurantoin). Also, compound **5f** exhibited good activity against the other three bacterial strains, with a MIC of 310μg/mL, much lower than those obtained for the reference drugs (Metronidazole, Nitrofurantoin). Compound **5b** which has the nitro group in position 4 of the pyrazole ring and where the benzimidazole ring is unsubstituted, as well as compounds **5c**, **5e** showed the best activity against *Staphylococcus aureus* strain ATCC259234 with a MIC of 190 μg/mL, much lower compared to the reference drugs (Metronidazole, Nitrofurantoin).

##### Antibiofilm Activity

The antibiofilm activity of the tested compounds has been quantified as the minimum inhibitory concentrations of bacterial biofilms (MBIC) expressed in μg/mL (Table 4, Figure 6). 

All **5a–g** synthesized compounds show better anti-biofilm activity against the *Staphylococcus aureus* strain ATCC259234 compared to the reference drugs. The most active compound, **5f**, was noted to inhibit a biofilm having an MBIC of 310 µg/mL against all the bacterial strains, much lower compared to the reference drugs (Metronidazole, Nitrofurantoin). Also, compound **5b** showed the best activity against *Staphylococcus aureus* strain ATCC259234 and *Pseudomonas aeruginosa* ATCC27853 with an MBIC of 190 μg/mL, much lower compared to the reference drugs (Metronidazole, Nitrofurantoin).

Generally, the antimicrobial activity results suggest that the synthesized compounds could represent promising leads for the development of broad-spectrum antibacterial activity and antibiofilm agents. 

#### 2.2.3. Mechanism of Action

The literature has proposed several mechanisms of action of antimicrobial agents depending on the type of drug. Taking into account that our compounds are containing bioactive groups found in other antimicrobial drugs, such as Metronidazole, we expect a similar mechanism of action, responsible for a microbicidal effect, for example, the penetration of the cellular membranes of anaerobic and aerobic pathogens, followed by inhibition of protein synthesis interaction with DNA, causing a loss of the helical structure of DNA and the breakdown of the chain [56]. On the other hand, Nitrofurantoin is taken up by bacterial intracellular nitroreductases to produce the active form of the drug by reducing the nitro group. The intermediate metabolites resulting from this reduction then bind to bacterial ribosomes and inhibit the bacterial enzymes involved in the synthesis of DNA, RNA, cell wall proteins, and other metabolic enzymes [57].

## 3. Materials and Methods

### 3.1. Chemistry

#### 3.1.1. General Chemical Characterization Techniques

The melting points were recorded on a Boetius hot plate and are uncorrected. TLC was carried out on silica gel Merck plates with methanol: dichloromethane:acetone = 2:6:2 as eluent and developed with a UV lamp (λ = 254 nm and 365 nm).

The spectra ^1^H-NMR and ^13^C-NMR spectra were recorded on a Varian Gemini 300BB operating at 300 MHz for ^1^H and 75 MHz for ^13^C in CDCl_3_ and the chemical shifts were relative to TMS as the internal standard.

A Varian Resolutions Pro Spectrometer was used to recordFTIR spectra using potassium bromide pellets.

A VSU-2P Zeiss-Jena Spectrophotometer, with MgO as a reference was used to obtain electronic spectra within the 400–800 nm range.

A Perkin Elmer 2400 Series II CHNS/O Elemental Analyzerwas used to determine elemental analyses and the results were in the range of ±0.4% of the theoretical value. Compounds **5a–g** have a purity >95%.

A triple quadrupole mass spectrometer model API3200 (Sciex) coupled with an Infiniti 1260 binary pump (Agilent) and autosampler was used for LC-MS/MS analyses. For data acquisition and processing, Analyst software version 1.5.2 was used. A Phenomenex Luna pentafluorophenyl (PFP2) column (100 × 2 mm, 3 μm, 100 Å) using a mobile phase composed of (*A*) water with 0.1% formic acid and (*B*) acetonitrile with 0.1% formic acid, at a flow rate of 0.25 mL/min was used to separate the samples. The injection volume was 10 µL. The mass spectrometer ESI interface was operated both in positive and negative ions mode (in different sample injections). Acquisitions were carried out in Q1 full scan over a mass/charge range from 50 to 700 Da.

#### 3.1.2. Mannich Bases **5a–g** Synthesis Method

Synthesis of compound **5e**. A mixture of 1-(hydroxymethyl)-3,5-dimethylpyrazole **2b** (0.0068 mol) and 1-amino-2-methylbenzimidazole (0.0068 mol) dissolved in methylene chloride was kept at room temperature for 50 for hours, with intermittent stirring. The progress of the reaction was monitored by TLC. After evaporation of the solvent, the Mannich base **5e** was obtained, which was then recrystallized from chloroform.

The synthesis of the other Mannich bases was carried out in a similar way.

##### Synthesis of N-[(1H-3,5-dimethylpyrazol-1-yl)methyl]-1-amino-1H-benzimidazole (**5a**)

Compound **5a** was synthesized according to the general procedure described for **5e**.

Yield 29%; mp 154–156 °C; R_f_ 0.18; ^1^H-NMR (300 MHz, CDCl_3_) δppm: 1.46 (s, 3H, 5-Me/pyrazole), 2.23 (s, 3H, 3-Me/pyrazole), 5.10 (bs, 2H, CH_2_N), 5.71 (s, 1H, H-4/pyrazole), 6.46 (bs, 1H, NH), 7.12 (s, 1H, H-2/benzimidazole), 7.17–7.25 (m, 3H, H-4, H-6, H-7/benzimidazole), 7.68–7.71 (m, 1H, H-5/benzimidazole). ^13^C-NMR (75 MHz, CDCl_3_) δppm: 9.83 (3-Me/pyrazole), 13.4 (5-Me/pyrazole), 63.0 (CH_2_N), 106.2 (C-4/pyrazole), 109.0 (C-7/benzimidazole), 120.1 (C-4/benzimidazole), 122.4, 123.5 (C-5, C-6/benzimidazole); 132.9, 140.8, 141.0 (C-5/pyrazole; C-3a, C-7a/benzimidazole), 142.5 (C-2/benzimidazole), 149.8 (C-3/pyrazole). IR (KBr, cm^−1^) 3335m *ν*(N-H), 1220vs *ν*(C_arom_-N), 1090s, *ν*(C_aliph_-N), 1556s, 1455s *ν*(benzimidazole ring) 1446m, 1324m*ν*(pyrazole ring). UV-Vis. (ethanol) *λ_max_* (log ε) **5a**: 322 (nm): 0.34, 277: 0.912, 269: 1.018, ESI-MS molecular ion [M+H]^+^: 242.201 Fragment ions: 118.200, 146.100, 129.100, 97.100, 91.100, 65.200; Anal. Calcd. for C_13_H_15_N_5_ (241.30): C 64.71; H 6.27; N 29.02; Found: C 64.37; H 6.67; N 28.71.

##### Synthesis of N-[(1H-3,5-dimethyl-4-nitropyrazol-1-yl)methyl]-1-amino-1H-benzimidazole (**5b**)

Compound **5b** was synthesized according to general procedure described for **5e**.

Yield 63%; mp 131–133 °C; R_f_ 0.78; ^1^H NMR (300 MHz, DMSO-d_6_) δppm: 2.10 (s, 3H, 5-Me), 2.34 (s, 3H, 3-Me), 5.40 (d, 2H, *J* = 5.2 Hz, CH_2_N), 6.31 (t, 1H, *J* = 5.2 Hz, NH), 7.14–7.25 (m, 3H, H-5, H-6, H-7/benzimidazole), 7.61–7.64 (m, 1H, H-4/benzimidazole), 7.92 (s, 1H, H-2/benzimidazole), 8.07 (t, 1H, *J* = 5.2 Hz, NH). ^13^C NMR (75 MHz, DMSO-d_6_) δppm: 10.4 (Me), 13.8 (Me), 64.5 (CH_2_N), 109.3 (C-7/benzimidazole), 119.3, 122.4, 122.7 (C-4, C-5, C-6/benzimidazole); 130.8 (C-4/pyrazole), 132.7 (C-3a/benzimidazole), 140.4, 141.2, 143.3, 145.3 (C-2, C-7a/benzimidazole, C-3, C-5/pyrazole). IR (KBr, cm^−1^) 3291vs *ν*(N-H), 1166s *ν*(C_arom_-N), 1096s *ν*(C_aliph_-N), 1566vs *ν*(NO_2_asymm), 1360vs *ν*(NO_2_symm),1455vs, 1342s *ν*(benzimidazole ring) 1442vs, 1306m *ν*(pyrazole ring). UV-Vis. (ethanol) *λ_max_* (log ε) **5b**: 278: 2.221 nm, ESI-MS molecular ion [M+H]^+^: 287.302 Fragment ions:118.200, 146.300, 129.200, 91.000;76.800, 65.100; Anal. Calcd. for C_13_H_14_N_6_O_2_ (286.29) C 54.54; H 4.93; N 29.35; O 11.18; Found: C 54.29; H 5.17; N 28.99.

##### Synthesis of N-[(1H-3,5-dimethyl-4-iodopyrazol-1yl)methyl]-1-amino-1H-benzimidazole (**5c**)

Compound **5c** was synthesized according to general procedure described for **5e**.

Yield 54%; mp 114–115 °C; R_f_ 0.50; ^1^H NMR (300 MHz, CDCl_3_) δppm: 1.58 (s, 3H, 5-Me), 2.15 (s, 3H, 3-Me), 5.22 (s, 2H, CH_2_N), 6.20 (bs, 1H, NH), 7.07 (d, 1H, *J* = 7.6 Hz, H-7/benzimidazole), 7.27–7.29 (m, 2H, H-5, H-6, benzimidazole), 7.34 (s, 1H, H-2/benzimidazole) 7.74 (d, 1H, *J* = 7.9 Hz, H-4/benzimidazole). ^13^C NMR (75 MHz, CDCl_3_) δppm: 11.1, 14.3 (2Me), 64.5, 64.7 (CH_2_N, C-4/pyrazole), 108.9 (C-7/benzimidazole), 120.5, 123.0, 123.9 (C-4, C-5, C-6/benzimidazole);133.0, 141.1 (C-4a, C-7a/benzimidazole),142.5 (C-5/pyrazole, C-2/benzimidazole),152.0 (C-3, pyrazole). IR (KBr, cm^−1^) 3300m *ν*(N-H), 1214vs *ν*(C_arom_-N), 1080vs *ν*(C_aliph_-N), 1490s, 1370s *ν*(benzimidazole ring), 1430m, 1361m *ν*(pyrazole ring). UV-Vis. (ethanol) *λ_max_* (log ε) **5c**: 328 (nm): 0.075, 275: 1.141, 268: 1.277. ESI-MS molecular ion [M+H]^+^: 368.104 Fragment ions: 118.200, 146.300, 129.200, 91.000, 65.100; Anal. Calcd. for C_13_H_14_IN_5_ (367.19) C 42.52; H 3.84; I 34.56; N 19.07; Found: C 42.13 H 4.21; N 19.33.

##### Synthesis of N-[(1H-pyrazol-1yl)methyl]-1-amino-2-methyl-1H-benzimidazole (**5d**)

Compound **5d** was synthesized according to general procedure described for **5e**.

Yield 87%; mp 144–146 °C; R_f_ 0.51; ^1^H NMR (300 MHz, CDCl_3_) δppm: 2.09 (s, 3H, 2-Me), 5.23 (d, 2H, *J* = 5.2 Hz, CH_2_N), 6.06–6.09 (m, 2H, NH, H-4/pyrazole), 6.67 (d, 1H, *J* = 7.9 Hz, H-7/benzimidazole), 6.85 (d, 1H, *J* = 2.1 Hz, H-5/pyrazole), 7.01, 7.10 (2t, 2H, *J* = 7.1 Hz, H-5, H-6/benzimidazole), 7.51–7.54 (m, 2H, H-4/benzimidazole, H-3/pyrazole). ^13^C NMR (75 MHz, CDCl_3_) δppm: 12.2 (Me), 66.3 (CH_2_N), 106.9 (C-4/pyrazole),108.1 (C-7/benzimidazole), 119.3 (C-4/benzimidazole), 122.2, 122.3 (C-5, C-6/benzimidazole);130.5 (C-5/pyrazole), 133.6, 140.4 (C-4a, C-7a/benzimidazole),141.5 (C-3/pyrazole),152.5 (C-2/benzimidazole). IR (KBr, cm^−1^) 3266m *ν*(N-H), 1281vs *ν*(C_arom_-N), 1066s *ν*(C_aliph_-N), 1529w, 1413w *ν*(benzimidazole ring), 1454w, 1343w *ν*(pyrazole ring). UV-Vis. (ethanol) *λ_max_* (log ε): 277: 0.548, 271: 0.549 nm, ESI-MS molecular ion [M+H]^+^: 228.001 Fragment ions: 132.300, 160.000, 104.000, 92.100, 65.200; Anal. Calcd. for C_12_H_13_N_5_ (227.27) C 63.42; H 5.77; N 30.82; Found: C 63.81; H 6.17; N 30.48.

##### Synthesis of N-[(1H-3,5-dimetyl-pyrazol-1-yl)methyl]-1-amino-2-methyl-1H-benzimidazole (**5e**)

Yield 61%; mp 99–100 °C; R_f_0.65; ^1^H NMR (300 MHz, CDCl_3_) δppm: 1.31 (s, 3H, 2-Me/pyrazole), 2.15 (s, 3H, Me), 2.17 (s, 3H, Me), 5.07 (d, 2H, *J* = 5.7 Hz, CH_2_N), 5.58 (s, 1H, H-4/pyrazole) 6.25 (t, 1H, *J* = 5.7 Hz, NH), 6.73–6.77 (m, 1H, H-7/benzimidazole), 6.98–7.04, 7.06–7.10 (2m, 2H, H-5, H-6/benzimidazole), 7.51–7.54 (m, 1H, H-4/benzimdazole). ^13^C-NMR (75 MHz, CDCl_3_) δppm: 9.63 (3-Me/pyrazole), 11.9 (s, 3H, Me), 13.4 (5-Me/pyrazole), 62.7 (CH_2_N), 106.1 (C-4/pyrazole), 107.7 (C-7/benzimidazole), 118.9 (C-4/benzimidazole), 122.0, 123.2 (C-5, C-6/benzimidazole); 133.4, 139.4, 140.3 (C-5/pyrazole; C-3a, C-7a/benzimidazole), 149.8 (C-3/pyrazole), 152.5 (C-2/benzimidazole). IR (KBr, cm^−1^) 3300m *ν*(N-H), 1232s *ν*(C_arom_-N), 1031s *ν*(C_aliph_-N), 1557m, 1425s *ν*(benzimidazole ring), 1457s, 1310m *ν*(pyrazole ring). UV-Vis. (ethanol) *λ_max_* (log ε): 276: 0.963, 271: 0.979 nm, ESI-MS molecular ion [M+H]^+^: 256.402 Fragment ions: 132.200, 160.100, 104.000, 91.900, 65.100; Anal. Calcd. for C_14_H_17_N_5_ (255.32) C 65.86; H 6.71; N 27.43; Found: C 65.7; H 7.06 N 27.81.

##### Synthesis of N-[(1H-3,5-dimethyl-4-nitropyrazol-1-yl)methyl]-1-amino-2-methyl-1H-benzimidazole (**5f**)

Compound **5f** was synthesized according to general procedure described for **5e**.

Yield 27%; mp 114–115 °C, R_f_ 0.67;^1^H NMR (300 MHz, CDCl_3_) δppm: 1.86 (s, 3H, Me), 2.29 (s, 3H, Me), 2.47 (s, 3H, Me), 5.21 (d, 2H, *J* = 5.2 Hz, CH_2_N), 6.20 (t, 1H, *J* = 5.2 Hz, NH), 6.69 (d, 1H, *J* = 7.8 Hz, H-7/benzimidazole), 7.03, 7.13 (2t, 2H, *J* = 7.4, 7.9 Hz, H-5, H-6/benzimidazole), 7.53 (d, 1H, *J* = 7.9 Hz, H-4/benzimidazole). ^13^C NMR (75 MHz, CDCl_3_) δppm: 12.4, 13.2, 14.2 (3Me), 63.8 (CH_2_N), 107.6 (C-7/benzimidazole), 119.4 (C-4/benzimidazole), 123.2, 123.3 (C-5, C-6/benzimidazole); 131.2 (C-4/pyrazole), 133.2, 141.8, 144.4 (C-3a, C-7a/benzimidazole, C-5/pyrazole) 147.6 (C-3/pyrazole), 151.9 (C-2/benzimidazole). IR (KBr, cm^−1^) 3300m*ν* (N-H), 1233m *ν*(C_arom_-N), 1056m *ν*(C_aliph_-N), 1565s *ν*(NO_2_asymm), 1359vs *ν*(NO_2_symm), 1489s, 1410m *ν*(benzimidazole ring), 1411s, 1334m *ν*(pyrazole ring). UV-Vis. (ethanol) *λ_max_* (log ε): 277: 0.921, ESI-MS molecular ion [M+H]^+^: 301.302 Fragment ions: 132.300, 160.200, 104.100, 92.100, 65.300; Anal. Calcd. for C_14_H_16_N_6_O_2_ (300.32) C 55.99; H 5.37; N; 27.98; O 10.65; Found: C 55.70; H 5.76; N 27.61.

##### Synthesis of N-[(1H-3,5-dimethyl-4-iodopyrazol-1yl)methyl]-1-amino-2-methyl-1H-benzimidazole (**5g**)

Compound **5g** was synthesized according to general procedure described for **5e**.

Yield 65%; mp 123–124 °C, R_f_ 0.57;^1^H NMR (300 MHz, CDCl_3_) δppm: 1.38 (s, 3H, Me), 2.17 (s, 3H, Me), 2.18 (s, 3H, Me),5.16 (d, 2H, *J* = 5.3 Hz, CH_2_N), 6.20 (t, 1H, *J* = 5.3 Hz, NH), 6.67 (d, 1H, *J* = 7.9 Hz, H-7/benzimidazole), 7.03, 7.12 (2t, 2H, *J* = 7.4 Hz, H-5, H-6/benzimidazole), 7.53 (d, 1H, *J* = 7.9 Hz, H-4/benzimidazole). ^13^C NMR (75 MHz, CDCl_3_) δppm: 10.8, 11.2, 14.0 (3Me), 64.0, 64.6 (CH_2_N, C-4/pyrazole), 108.1 (C-7/benzimidazole), 119.3 (C-4/benzimidazole), 122.8 (C-5, C-6/benzimidazole);133.5/140.4 (C-3a, C-7a, benzimidazole),142.4 (C-5/pyrazole), 151.4 (C-3/pyrazole),151.9 (C-2, benzimidazole). IR (KBr, cm^−1^) 3300m *ν*(N-H), 1208vs *ν*(C_arom_-N), 1034s *ν*(C_aliph_-N), 1539s, 1404s *ν*(benzimidazole ring), 1437s, 1308s *ν*(pyrazole ring). UV-Vis. (ethanol) *λ_max_* (log ε): 341: 0.14, 275: 2.11, 268: 2.23 nm, ESI-MS molecular ion [M+H]^+^: 382.604 Fragment ions: 132.100, 160.100, 104.200. 92.100, 65.300; Anal. Calcd. for C_14_H_16_IN_5_ (381.22) C 44.11; H 4.23; I 33.29; N 18.37; Found: C 44.48; H 4.63; N 18.01.

### 3.2. Biological Tests

#### 3.2.1. Cytotoxicity of Samples

The reagents used for the in vitro cytotoxicity test were supplied by Merck.

To evaluate the biocompatibility of the compounds, a stabilized mouse fibroblast cell line NCTC (clone 929) cultured in minimal essential medium (MEM) containing 10% fetal bovine serum (FBS) and 2 mM L-glutamine, 100 U/mL penicillin, 100 µg/mL streptomycin and 500 µg/mL neomycin was used.

For new pyrazolo-benzimidazole derivatives testing, the compounds were solubilized in 100 μL DMSO. The solutions of the compounds were used in the cytotoxicity experiments at concentrations: 0.001 μM, 0.01 μM, 0.1 μM, 1 μM, 0.1 mM, and 1 mM.

The NCTC cell suspension was seeded in 96-well culture plates at a density of 4×104 cells/mL and incubated in a 5% CO_2_ humidified atmosphere at 37 °C for 24 h.

The studied compounds were added in triplicate at different concentrations from 0.001 μM to 1 mM to each well and plates were incubated in the standard conditions, for 48 h.Cell viability was assessed using the MTT method.

Afterward, the MTT solution was replaced with isopropanol to dissolve the formazan crystals. The absorbance of the resulting colored solution was then measured at 570 nm using a Berthold Mithras LB 940 microplate reader (Germany). The results were calculated according to Equation (1) knowing that the measured optical density is directly proportional to the number of viable cells present in the tested cell culture.
(1)% cell viability=(OD sample/OD control) × 100%

Untreated cells were used as control, considered 100% viable cells.

#### 3.2.2. Cell Morphology Examination

For morphology examination of cells cultivated in the presence of compounds, fibroblasts were fixed with methanol and stained with Giemsa. The cultures were visualized by light microscopy using an optic microscope Carl Zeiss Axio Observer D1 (Germany); images were taken with the digital camera Axio Cam MRc (Germany).

#### 3.2.3. Statistical Analysis

Data were expressed as mean value ± SD for three independent samples (n = 3). Statistical analysis of the data was performed using the one-tailed paired Student’s t-test, on each pair of interest. Differences were considered statistically significant at *p* < 0.05, as a minimal level of significance.

#### 3.2.4. Quantitative Testing of Antimicrobial Activity on Bacterial Strains

Quantitative testing was performed by the method of serial microdilutions in liquid medium TSB (Tryptone Soy Broth) using 96-well plates, in order to determine the minimum inhibitory concentration CMI (minimum inhibitory concentration), the minimum amount of chemical compound capable of inhibiting cell growth microbial. In a volume of 90 μL of culture medium, binary serial dilutions of the stock solution of the compound prepared in DMSO (10 µg/mL) were performed. Here, 90 μL of liquid culture medium and 90 μL of the chemical compound were pipetted into the first well. Then, 90 μL was transferred from the first well to the second, 90 μL was transferred from the second well to the third, and so on to the last well, from which 90 μL was discarded. Subsequently, the wells were seeded with 10 µL of microbial suspension with a density of 106 CFU (Colony Forming Units). The microbial suspensions were made in AFS (sterile physiological water) from 24-h cultures obtained on simple agar. Each test also worked with a microbial culture control (a series of wells containing exclusively culture medium inoculated with microbial suspension) and an environmental sterility control. After incubating the plates at 37 °C for 24 h, the results obtained by macroscopic examination and absorbance reading at 620 nm were analyzed. In the growth control well, the environment was cloudy due to microbial growth. The obligatory sterility control well did not show any bacterial growth, the liquid content remained clear and transparent. The concentration of the chemical compound corresponding to the last well in which no culture development was observed was the MIC value (mg/mL) for that compound.

#### 3.2.5. Antibiofilm Assay

Microbial cells were cultured in 96-well plates with nutrient broth and in the presence of test compounds (following MIC readings), they were incubated at 37 °C for 24 h. The plates were emptied and washed three times with AFS. This was followed by fixation of the adhered cells with 110 μL of methanol for 5 min, after which the methanol solution was removed by inversion. The adhered cells were stained with 1% purple crystal alkaline solution (110 μL/well) for 15 min after staining, the solution was removed, then the plates were washed under running tap water. The microbial biofilms formed on the plastic plates were resuspended in 33% acetic acid (by bubbling), and the intensity of the colored suspension was assessed by reading the absorbance at 492 nm.

## 4. Conclusions

In the presented work, according to the synthesis strategy, new pyrazolo-benzimiazole hybrids were successfully obtained. It has been demonstrated that the HOSA reagent is useful in creating new N-N bonds of the hydrazine type. The structures of the new synthesized hybrids were confirmed by assigning NMR, MS, FTIR, UV-VIS, and elemental analysis spectra.

In vitro cytotoxicity evaluation of the new compounds demonstrated a good correlation between morphological observations and quantitative results of cell viability. Both methods indicated that the new synthesized pyrazolo-benzimidazole hybrids were noncytotoxic up to a concentration of 1 μM in the studied concentration range.

The results obtained by the antimicrobial testing of the newly synthesized hybrids showed that these compounds have superior antimicrobial activity to the mononuclear heterocycles from which they originate.

All synthesized **5a–g** compounds with a few exceptions are more active against all bacterial tested strains (Gram-positive strains *Staphylococcus aureus* ATCC25923, *Enterococcus faecalis* ATCC29212,and Gram-negative strains *Pseudomonas aeruginosa* ATCC27853, *Escherichia coli* ATCC25922) than the reference drugs (Metronidazole, Nitrofurantoin).

The most active compounds that have good antibacterial activity and inhibit biofilm are compounds **5b** and **5f**, which contain a nitro group grafted on the pyrazole ring, which differs in the nature of the methyl substituent grafted on the benzimidazole ring.

Compound **5f** stands out as the most active against *Staphylococcus aureus* ATCC 25923, with a MIC of 150 μg/mL, and against the other three bacterial strains, with a MIC of 310 μg/mL compared to reference drugs (Metronidazole, Nitrofurantoin).

Also, compound **5f** was noted to inhibit the biofilm having an MBIC of 310 µg/mL against all the bacterial strains versus the reference drugs.

It can be concluded that generally synthesized Mannich bases have good antimicrobial activity and inhibit biofilm, therefore they are good candidates for therapeutic purposes.

## Data Availability

The data presented in this study are available on request from the corresponding author.

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
