# Peer review of "New Pyrazolo-Benzimidazole Mannich Bases with Antimicrobial and Antibiofilm Activities"

_antibiotics, 2022, doi:10.3390/antibiotics11081094_

Round 1

Reviewer 1 Report

This paper presents some useful results through the synthesis of some relatively simple new Mannich bases, many of which show more potent biological activity than existing standards.  These results are therefore acceptable for publication.  Prior to this, I suggest some modification to the manuscript.  In particular, the reaction mechanisms shown in Schemes 4 and 5 are well known and understood and consequently should be deleted as they are unnecessary.  In Scheme 1, pyrazole should be indicated by 1a to which it is referred in the text.  As an option, I suggest putting compounds 5a-g in a Table for the sake of clarity, as compound structures are not always clear from the Schemes, but only from the Experimental details.  The general English text needs some moderate correction, and this includes spacing between sentences.

Author Response

Thank you for reviewing our article:
"New pyrazolo-benzimidazole Mannich bases with antimicrobial and antibiofilm activities" and for the observations and useful recommendations received.
This paper presents some useful results through the synthesis of some relatively simple new Mannich bases, many of which show more potent biological activity than existing standards. These results are therefore acceptable for publication. Prior to this, I suggest some modification to the manuscript.
The general English text needs some moderate correction, and this includes spacing between sentences.
1. In particular, the reaction mechanisms shown in Schemes 4 and 5 are well known and understood and consequently should be deleted as they are unnecessary.
Thank you for your observation. That's right, the mechanism is known, that's why only a short comment on the development of the reaction is necessary.
2. In particular, the reaction mechanisms shown in Schemes 4 and 5 are well known and understood and consequently should be deleted as they are unnecessary.
Instead of schemes 4a and 4b we wrote:
“The substituted benzimidazoles 3a,b were obtained according to published methods [50] with good yields of 75%, respectively 48%, through the nucleophilic attack of 1,2-phenylenediamine on the intermediate carboniu, formed in an acidic medium from the corresponding carboxylic acid.”
3. In Scheme 1, pyrazole should be indicated by 1a to which it is referred in the text.
We wrote the structure of the pyrazole 1a, which is included in Scheme 1.
4. As an option, I suggest putting compounds 5a-g in a Table for the sake of clarity, as compound structures are not always clear from the Schemes, but only from the Experimental details.
5. The general English text needs some moderate correction, and this includes spacing between sentences.
The spacing between words has been corrected.
I was careful to space the words in the text, but moving the text from one laptop to another can join some words.
That is why I am also sending you the revised manuscript in pdf format.

Reviewer 2 Report

The manuscript entitled “New pyrazolo-benzimidazole Mannich bases with antimicrobial and antibiofilm activities” by Zălaru et al. reports the synthesis, characterization, and antibacterial and antibiofilm activity analysis of a new series pyrazolo-benzimidazole hybrid Mannich bases. While the study is interesting in terms of the development of the hybrid compound and its characterization using NMR, IR, UV-Vis, MS, and elemental analyses; to make the study more significant, biological part of the study should also be strong. The authors reported the antibacterial and antibiofilm activities of the hybrid compound against several pathogenic micro-organisms, but authors have not reported the toxicity of these compounds in the host cells, which is important when the compounds have to be used for therapeutic purposes.

Authors must include the experiment for toxicity assays in the host cells. Because the MIC of the compound seems quite high that can affect the host cells.

Author Response

We agree that in order to make the study more significant, it is necessary to determine the cytotoxicity. The article also has cytotoxicity results.

Reviewer 3 Report

In this article the authors evaluate the antimicrobial and antibiofilm activities of several pyrazolo-benzimidazole Mannich bases against several strains. The studies performed showed that some compounds had good antimicrobial activity and inhibit biofilm.

The manuscript is written in fluent style and there seem to be no significative spelling mistakes. This contribution may lead to good candidates for therapeutic agents in the future. In the article they synthesized and evaluate 7 pyrazolo-benzoimidazole adducts.

There are some issues which I would like to address:

-In line 39 the sentence “A number of clinically tested drugs have been reported in the literature, including (…)” should refer to the moiety the authors are going to discuss later. “A number of clinically tested drugs, which contains a pyrazole ring, have been reported in the literature, including (…)”

-In line 48 where it says “activity” it would be better to put “activities” as they later put some bioactivities (“such as antitumor, antiproliferative, anticancer, local anesthetics (…)”).

-In line 53 where it says “attached” (“benzimidazoles attached to other heterocyclic moieties”) where authors referring to hybrids of benzimidazoles and another heterocyclic compound? Because the way it is stated it seems that they are referring to some sort of fused polycyclic compound. If so, maybe it should be better to put “benzimidazoles linked to other heterocyclic moieties”

-Probably the part of the benzimidazole-core drugs (from line 59 to line 64) should be better placed before the part referring to the hybrids (from line 53 to line 58).

-In line 71 when authors state that “(…) we decided to synthesize a new series of benzimidazole derivatives incorporating a pyrazole ring adapting a fragment-based drug design strategy, with the aim of obtaining molecules with improved antibacterial activity.” Are they following a design synthesis/compounds focused in a particular target? Are the moieties used (pyrazole and benzimidazole) selected for some reason (in the introduction their bioactivities are reported but there doesn’t seem to have any remarkable antibacterial activity).

-In line 86 it is stated that “pyrazole 1a is commercially available” but the structure doesn’t appear. Is the molecule in scheme 1 that doesn’t have numeration (the one from which compound 2a is obtained)?

-Through the article in many places (as in lines 88-89, 114,…) yields below 70% are referred to as “good yields”. Usually yield of 100% are quantitative, above 90% excellent, above 80% very good, above 70% good, above 50% fair/moderate and below 40% poor.

-In scheme 1 the reaction conditions (lines 94-95) should be reviewed, (ii) is missing.

-In line 98 where it says “with good yields of 75%, respectively 48%.· it should say “with yields of 75% and 48% respectively” The same with line 101 (“with good yields 62%, respectively 63%.”).

-In line 107 (scheme 2), the reaction conditions of iv are missing.

-In the mechanism of the reaction, scheme 4a is not necessary. Also, in scheme 4b, in the last compound the “R” is missing.

-The settings of all molecules should be checked (some of them have odd angles and bond lengths).

-In lines 154-155 it says that “thin layer chromatography confirmed the purity of the new synthesized 154 compounds” but in the experimental section compounds 5a-5f have elemental analysis, which is better to determine the purity (as well as HPLC). Also, the NMR1H given in the supplementary material (specially the ones of 5b, 5c and 5e) don’t have good resolution and they don’t seem to be very pure. If there are issues with the NMR maybe an HPLC should address the purity.

-In line 207 where it says “Figure 6” where authors referring to “Scheme 6”?

-In scheme 6 the substituent “R” and “R1” are always the same (R=H and R1=CH3), so it is better if they are removed.

-Regarding the bioactivities: In tables 1 and 2, the SEM deviations of the MIC are missing. Also, it should state the number of times the experiments were performed. It would be interesting to place the MIC values ​​in µM. Also figure 4 seems to give the same information as table 1, the same with figure 5 and table 2. Maybe figures 4 and 5 could be moved to the supplementary material section.

-Regarding the SAR (lines 239-254): there are few compounds for a SAR. Also, the discussion is just a description of the substituents in each molecule. The reason why these substituents were selected and the way they influence the activity are not discussed.

-Regarding the mechanism of action (lines 275-285): The mechanisms discussed are those of the reference drugs metronidazole and nitrofurantoin, but there is not discussion of the synthesized compounds. Are the authors implying that they could have similar mechanisms of action to those of the reference drugs?

-Regarding the Mannich basis 5a-g synthesis method (line 11), the general method for the synthesis of these compounds isn’t really a general one. Where is says “A mixture of 1-(hydroxymethyl)-3,5-dimethylpyrazole 2b (0.0068 mol) and …” it should say “A mixture of the corresponding pyrazole (0.0068 mol) and…”, and in the end (line 315) remove the 5e reference (Mannich base 5e was obtained and recrystallized from chloroform.).

-If the signals of NMR1H and NMR13C are going to be assigned they should be all of them assigned (not just some). Were bidimensional experiments used in order to do the assignment? Also, the way of given the data is a little bit complicated and when two signals have the same multiplicity but different ppm, the description of it should be placed after each signal not altogether. For example, in line 408-409 “1.86, 2.29, 2.47 (3s, 9H, 3Me)” should be “1.86 (s, 3H, Me), 2.29 (s, 3H, Me), 2.47 (s, 3H, Me)” because as it is written it confusing because of the integral value.

-Please note that in NMR1H of compound 5b there is an extra hydrogen (there seem to be 15H and the molecule has 14).

-In the NMR13C of compound 5e there is an extra carbon and in compound 5g the is one carbon missing (or is it the one at 122.8 for two carbons?).

-The ESI-MS only has one decimal? Usually it gives four, the experimental and theorical values and the molecular formula from which is calculated.

-The conclusions should be rewritten because they feel like just a summary of what has been done.

Kind regards

Author Response

In this article the authors evaluate the antimicrobial and antibiofilm activities of several pyrazolo-benzimidazole Mannich bases against several strains. The studies performed showed that some compounds had good antimicrobial activity and inhibit biofilm.
The manuscript is written in fluent style and there seem to be no significative spelling mistakes. This contribution may lead to good candidates for therapeutic agents in the future. In the article they synthesized and evaluate 7 pyrazolo-benzoimidazole adducts.
There are some issues which I would like to address:
-In line 39 the sentence “A number of clinically tested drugs have been reported in the literature, including (…)” should refer to the moiety the authors are going to discuss later. “A number of clinically tested drugs, which contains a pyrazole ring, have been reported in the literature, including (…)” A number of clinically tested pyrazole-containing drugs have been reported in the literature, including Lonazolac (a), new line 55
-In line 48 where it says “activity” it would be better to put “activities” as they later put some bioactivities (“such as antitumor, antiproliferative, anticancer, local anesthetics (…)”). We corrected it “remarkable biological activities” new line 64
-In line 53 where it says “attached” (“benzimidazoles attached to other heterocyclic moieties”) where authors referring to hybrids of benzimidazoles and another heterocyclic compound? Because the way it is stated it seems that they are referring to some sort of fused polycyclic compound. If so, maybe it should be better to put “benzimidazoles linked to other heterocyclic moieties”
In this case we refer here to" benzimidazoles attached to other heterocyclic moieties" it is correct.
-Probably the part of the benzimidazole-core drugs (from line 59 to line 64) should be better placed before the part referring to the hybrids (from line 53 to line 58). New line 69 to line 73
It was done.
-In line 71 when authors state that “(…) we decided to synthesize a new series of benzimidazole derivatives incorporating a pyrazole ring adapting a fragment-based drug design strategy, with the aim of obtaining molecules with improved antibacterial activity.” Are they following a design synthesis/compounds focused in a particular target? Are the moieties used (pyrazole and benzimidazole) selected for some reason (in the introduction their bioactivities are reported but there doesn’t seem to have any remarkable antibacterial activity).
We wrote:
"In our previous study [42] we synthesized alkylaminopyrazoles with a suitable alkyl chain to have antimicrobial activity, and in addition, an supplementary pharmacophore pyrazole ring. Starting from the previous results [42], we created the design of new molecules, by substituting the alkyl chain with another pharmacophore ring, resulting in fact a pyrazolo-benzimidazole hybrid, in order to obtain molecules with improved antimicrobial activity". New lines 89 to 93.
-In line 86 it is stated that “pyrazole 1a is commercially available” but the structure doesn’t appear. Is the molecule in scheme 1 that doesn’t have numeration (the one from which compound 2a is obtained)? We wrote the structure of the pyrazole 1a, which is included in Scheme 1. New line 113.
-Through the article in many places (as in lines 88-89, 114,…) yields below 70% are referred to as “good yields”. Usually yield of 100% are quantitative, above 90% excellent, above 80% very good, above 70% good, above 50% fair/moderate and below 40% poor. We revised in text these corrections about yields. New lines 119 to 123.
-In scheme 1 the reaction conditions (lines 94-95) should be reviewed, (ii) is missing. We added in Scheme 1 the reaction conditions. New lines 115-116.
-In line 98 where it says “with good yields of 75%, respectively 48%.· it should say “with yields of 75% and 48% respectively” The same with line 101 (“with good yields 62%, respectively 63%.”). We modified in the text. New lines 119-120.
-In line 107 (scheme 2), the reaction conditions of iv are missing. We added in text the reaction conditions. New lines 130-132.
-In the mechanism of the reaction, scheme 4a is not necessary. Also, in scheme 4b, in the last compound the “R” is missing.
The schemes mentioned above have been deleted.
-The settings of all molecules should be checked (some of them have odd angles and bond lengths).
This is no longer the case, schemes 4a and 4b have been deleted.
-In lines 154-155 it says that “thin layer chromatography confirmed the purity of the new synthesized 154 compounds” but in the experimental section compounds 5a-5f have elemental analysis, which is better to determine the purity (as well as HPLC). Also, the NMR1H given in the supplementary material (specially the ones of 5b, 5c and 5e) don’t have good resolution and they don’t seem to be very pure. If there are issues with the NMR maybe an HPLC should address the purity.
The phrase remains in the article.
"Also, thin layer chromatography confirmed the purity of the new synthesized compounds (by the values of the Rf retention indices given in the Experimental section)." We wrote "The values of the Rf retention indices are given in the Experimental section." New line 165.
-In line 207 where it says “Figure 6” where authors referring to “Scheme 6”? Yes, We refer to schemes 6 and 7 and not Figure 6. We corrected. New line 217.
-In scheme 6 the substituent “R” and “R1” are always the same (R=H and R1=CH3), so it is better if they are removed. We wrote on the scheme R = H and R1 = CH3. New lines 225-231.
-Regarding the bioactivities: In tables 1 and 2, the SEM deviations of the MIC are missing.
We put the standard deviations.
-Also, it should state the number of times the experiments were performed. It would be interesting to place the MIC values in μM.
The values of MIC are in μM.
-Also figure 4 seems to give the same information as table 1, the same with figure 5 and table 2. Maybe figures 4 and 5 could be moved to the supplementary material section.
Figures 4 and 5 are moved to the supplementary material.
-Regarding the SAR (lines 239-254): there are few compounds for a SAR. Also, the discussion is just a description of the substituents in each molecule. The reason why these substituents were selected and the way they influence the activity are not discussed.
We did not refer to SAR, but only a correlation between the substituents present in the molecule and the corresponding antimicrobial activity. New lines 302-303.
-Regarding the mechanism of action (lines 275-285): The mechanisms discussed are those of the reference drugs metronidazole and nitrofurantoin, but there is not discussion of the synthesized compounds. Are the authors implying that they could have similar mechanisms of action to those of the reference drugs?
The discussed mechanisms are those of the reference drugs, Metronidazole and Nitrofurantoin, but we cannot say that the action mechanisms of our compounds are similar to those of the references. New lines 339-350.
-Regarding the Mannich basis 5a-g synthesis method (line 11), the general method for the synthesis of these compounds isn’t really a general one. Where is says “A mixture of 1-(hydroxymethyl)-3,5-dimethylpyrazole 2b (0.0068 mol) and …” it should say “A mixture of the corresponding pyrazole (0.0068 mol) and…”, and in the end (line 315) remove the 5e reference (Mannich base 5e was obtained and recrystallized from chloroform.).
To be as clear as possible, we chose to write the synthesis method for obtaining compound 5e and the others are obtained in a similar way. New lines 378-382.
-If the signals of NMR1H and NMR13C are going to be assigned they should be all of them assigned (not just some). Were bidimensional experiments used in order to do the assignment? Also, the way of given the data is a little bit complicated and when two signals have the same multiplicity but different ppm, the description of it should be placed after each signal not altogether. For example, in line 408-409 “1.86, 2.29, 2.47 (3s, 9H, 3Me)” should be “1.86 (s, 3H, Me), 2.29 (s, 3H, Me), 2.47 (s, 3H, Me)” because as it is written it confusing because of the integral value.
- We corrected the interpretation of 1H-RMN signals.
-Please note that in NMR1H of compound 5b there is an extra hydrogen (there seem to be 15H and the molecule has 14).
-In the NMR13C of compound 5e there is an extra carbon and in compound 5g the is one carbon missing (or is it the one at 122.8 for two carbons?).
To eliminate ambiguities in the NMR spectra of compounds 5b, 5e and 5g, these spectra were recorded again.
-The ESI-MS only has one decimal? Usually it gives four, the experimental and theorical values and the molecular formula from which is calculated.
"Four decimals mass units are reported when a high resolution analyser is used. In this case, the ESI mass spectra were collected on a triple quadrupole instrument (as reported in the experimental section), that is not high resolution. This is acceptable as long as NMR of the same compounds is carried out, too."
We provided the ESI-MS data with three decimal places.
-The conclusions should be rewritten because they feel like just a summary of what has been done.

Round 2

Reviewer 2 Report

The manuscript entitled “New pyrazolo-benzimidazole Mannich bases with antimicrobial and antibiofilm activities” by Zălaru et al. reports the synthesis, characterization, and antibacterial and antibiofilm activity analysis of a new series pyrazolo-benzimidazole hybrid Mannich bases. The author have incorporated the cytotoxicity assay using the hybrid molecule as asked in the revision. The manuscript is now seem to be suitable for publication.

Author Response

Thank you for your kindness in reviewing the results obtained following the determination of cytotoxicity, also for the approval and the verdict given for publication.
